# Seroprevalence Study of Pertussis in Adults at Childbearing Age and Young Infants Reveals the Necessity of Booster Immunizations in Adults in China

**DOI:** 10.3390/vaccines10010084

**Published:** 2022-01-06

**Authors:** Zhiyun Chen, Jie Pang, Nan Zhang, Ning Chen, Yiwei Ding, Qiushui He

**Affiliations:** 1Department of Medical Microbiology, Capital Medical University, Beijing 100069, China; chenzy@ccmu.edu.cn (Z.C.); 122019000048@ccmu.edu.cn (J.P.); nan@ccmu.edu.cn (N.Z.); chenning@ccmu.edu.cn (N.C.); dingyiwei@ccmu.edu.cn (Y.D.); 2The Sixth Medical Center, Department of Respiratory Medicine, Chinese PLA General l Hospital, Beijing 100048, China; 3Research Center for Infections and Immunity, Institute of Biomedicine, University of Turku, 20520 Turku, Finland

**Keywords:** pertussis, anti-PT IgG antibodies, seroprevalence, infant, adults, China

## Abstract

In China, the vaccination strategy against pertussis is started from 3 months of age, with no booster dose used after the booster given at two years. Despite a high vaccination coverage, pertussis has been increasingly reported since the last decade. This study evaluates the prevalence of serum anti-pertussis toxin (PT) IgG antibodies in adults at childbearing age and infants before the age of primary immunization in Beijing, China. A total of 1175 serum samples randomly selected from individuals who attended an annual health examination at the Sixth Medical Center of the PLA General Hospital, Beijing, in 2019, was included. The geometric mean concentration (GMC) and median concentration of anti-PT IgG antibodies among adults aged 20–39 years were 3.81 IU/mL and 3.24 IU/mL, and the corresponding concentrations were 1.72 IU/mL and 1.43 IU/mL among infants under 3 months of age. The seroprevalence of PT IgG antibodies ≥ 40 IU/mL in adults and infants was 2.0% (15/735) and 1.1% (5/440). In total, 65.99% (485/735) of adults and 83.41% (367/440) of infants had non-detectable pertussis-specific antibodies (<5 IU/mL). Our results showed that the majority of adults at a reproductive age and young infants are vulnerable to pertussis, suggesting that booster vaccinations in adults should be considered in this country.

## 1. Introduction

Whooping cough, or pertussis, is an acute respiratory infection and is mainly caused by the Gram-negative bacillus *Bordetella pertussis*. Infants too young to be fully vaccinated are at the highest risk of contracting pertussis, and hospitalizations and mortality rates are high in infants under 3 months of age [1]. In China, whole-cell pertussis vaccines (WPVs) were introduced in the 1960s. Since the implementation of the national immunization program in 1978, the incidence and mortality of pertussis have decreased significantly. A combined diphtheria–tetanus–acellular pertussis (DTaP) vaccine has been in use from 2007, and completely replaced the diphtheria–tetanus–whole pertussis (DTwP) vaccine by 2013. However, over the following few years, the incidence of pertussis has continued to rise (Figure 1) [2,3]. The Chinese vaccination strategy is primarily administered with three doses of DTaP vaccines at the age of 3, 4, and 5 months, with a booster dose given at 18–24 months. After the booster dose at two years of age, no booster dose is given. This vaccination strategy has remained unchanged since the 1980s. Therefore, most adults at childbearing age and young infants may not have enough protective antibodies such as anti-pertussis toxin (PT) IgG antibodies. A recent cross-sectional study conducted in Beijing, China, compared the seroprevalence of anti-PT IgG antibodies in samples collected during two study periods (2010 and 2015/2016) and found that the seroprevalence of anti-PT IgG antibodies (≥40 IU/mL indicative of a pertussis infection within past two years) was 5.1% in 2010 and 4.0% in 2015/2016. Moreover, the rate of undetectable anti-PT IgG antibodies (<5 IU/mL) has significantly increased between the two periods [4]. It is well known that parents and other family members are the main source of infection for infants and young children [5,6]. Newborns are mainly protected from maternal antibodies [7,8]. The concentrations of anti-pertussis antibodies in cord blood samples are equal to or even higher than those observed in the mothers. However, Meng et al. compared 194 paired maternal and cord blood samples collected in Beijing from 2016 to 2017 and found that most pregnant women (70.1%) and newborns (74.8%) were generally lacking protective antibodies to pertussis [9]. In general, data on the occurrence of pertussis between adults at childbearing age and young infants are quite limited. Even in the above-mentioned studies, the number of subjects was not sizable.

In this present study, we aim to evaluate the level of serum anti-PT IgG antibodies in adults at childbearing age and infants under 3 months of age in Beijing to better understand the seroepidemiology of pertussis in these populations, and to provide important information on how effective the current immunization strategy in China is.

## 2. Materials and Methods

### 2.1. Study Subjects and Serum Samples

Serum samples were collected from adults and infants who attended annual health examinations at the Sixth Medical Center of People Liberation Army General Hospital, Beijing, in 2019. Altogether, 1631 sera from adults at a childbearing age (20–39 years old) and 440 sera from infants under 3 months of age were collected. Of these, 735 (45.06%) sera from adults and 440 (100%) sera from infants were included in this study. The only information collected from the study subjects was age, gender, and date of sampling. The information on the vaccination status of these subjects was not collected. Because the first dose of DTaP vaccine is given at 3 months of age, all infants were considered unvaccinated. Since pertussis vaccination was introduced in 1960s, many adults should have received pertussis vaccines.

### 2.2. Serological Testing

Commercial ELISA kits (Institut Virion/Serion GmbH, Würzburg, Germany) were used to determine concentration of anti-PT IgG antibodies according to the manufacturer’s instructions. The interpretation of the results was described previously [10]. Antibodies greater than or equal to 100 IU/mL indicated a recent infection within one year, and a value between 40 and 100 IU/mL indicated an infection within the past few years. When the concentration values were below 5 IU/mL, it was considered that no antibodies were detected. When the concentration was between 5 and 40 IU/mL, it was considered seronegative.

### 2.3. Statistical Analysis

Data were analyzed using the GraphPad Prism 7 version (San Diego, CA, USA) and SPSS version 25.0 (SPSS Inc., Chicago, IL, USA). Serum anti-PT IgG concentrations in different age groups and genders were analyzed by a normality test. Normally distributed continuous variables of the two groups were compared using a Student’s t-test. Non-normally distributed continuous variables of the two groups were compared using the Mann–Whitney U test. Seroprevalence and the proportion of subjects with non-detectable anti-PT IgG between adults and infants were compared by the Chi-square test. Two-tailed *p* values < 0.05 were considered statistically significant. 

## 3. Results

A total of 1175 serum samples was included in this study. Of them, 735 included 328 females and 407 males aged 20–39 years, and 440 included 218 females and 222 males aged under 3 months old (Table 1). Among the infant samples, 248 cases were one month old, and 192 cases were two months old. 

The geometric mean concentration (GMC) and median concentration of anti-PT IgG antibodies among adult subjects were 3.81 IU/mL and 3.24 IU/mL. No difference was found between men and women. Altogether, 15 (2.0%) subjects had anti-PT IgG antibodies higher than 40 IU/mL. There was no statistical difference between men and women (*p* = 0.37) (Table 1). Among these 15 subjects, 14 (1.9%), including 9 men and 4 women, had a concentration between about 40 and 100 IU/mL, and 1 (0.1%) had a concentration ≥ 100 IU/mL. The non-detectable rate of anti-PT IgG antibodies was 66.0% (Figure 2). No statistical difference in the prevalence of non-detectable anti-PT IgG antibodies was noticed between men and women (*p* = 0.48).

Among 440 infants, the GMC and median concentrations of serum anti-PT IgG antibodies were 1.72 IU/mL and 1.43 IU/mL (Table 1). There were five (1.1%) subjects who had antibodies higher than 40 IU/mL. Among these five subjects, four (0.9%) had a concentration between about 40 and 100 IU/mL, and one (0.2%) had a concentration ≥ 100 IU/mL. There were 367 (83.4%) infants who had non-detectable anti-PT IgG antibodies (Figure 2). The GMC and median concentration of anti-PT IgG antibodies were 1.73 IU/mL and 1.43 IU/mL among one-month-old infants, and 1.71 IU/mL and 3.17 IU/mL among two-month-old infants. However, no differences were observed between the two groups (*p* = 0.56) (Figure 3).

Although there was no difference in the level of serum anti-PT IgG antibodies between groups of infants and adults (*p* = 0.25), the non-detectable rate of anti-PT IgG antibodies was significantly higher in infants than in adults (83.4% vs. 66.0%, *p* < 0.001).

## 4. Discussion

Pertussis is still a significant public health problem throughout the world. It is well known that pertussis is no longer just a childhood disease. Many studies have reported the occurrence of adult pertussis [4,11,12,13]. Typical clinical characteristics of pertussis include a paroxysmal cough and whooping. In adults, however, the typical symptoms of pertussis are not often present, and atypical symptoms such as a persistent cough are not uncommon [14,15]. Due to the increase in atypical clinical cases and asymptomatic infections, as with many countries, the incidence of pertussis in China is also most likely under-reported [16,17,18,19]. Similarly, we also found the seroprevalence of 2.0% in adults aged 20~39 years old in our study. It has been more than 18 years since they were vaccinated. Therefore, those adults who are at a childbearing age who had antibodies ≥ 40 IU/mL were considered to have a real *B. pertussis* infection and are probably the main source of infant pertussis.

In this study, we found that, in 2019, 66% of adults at a childbearing age and 83.4% of young infants before the age of the first dose of pertussis vaccination did not have detectable anti-PT IgG antibodies, showing that they were vulnerable to pertussis. This finding was in line with a recent study carried out in Beijing during the period of 2016–2017, in which 70.1% (95% CI: 63.6–76.1%) of mothers and 74.8% (95% CI: 68.2–80.3%) of newborns did not have detectable serum anti-PT IgG antibodies [9].

Several earlier studies have shown that the level of anti-PT IgG antibodies in cord bloods is close to or higher than that of maternal antibodies. The level gradually decreases in the first few months after birth, and the antibody concentration becomes negligible by 2–4 months [20,21]. In a recent study, the half-life of pertussis-specific antibodies in infants induced by the maternal Tdap vaccination (29–36 days) was shorter than previously reported data in the pre-Tdap era (5–6 weeks) [22]. In our study, although 1.1% of infants under three months of age had positive PT IgG antibodies, we could not differentiate whether they had a real pertussis infection, or if they were from maternal antibodies. No statistical difference was found in the level of antibody concentrations between one-month-old and two-month-old infants. In addition, we did not observe differences between men and women.

Vaccination is the most effective way to prevent pertussis. Although the percentages of DTaP vaccination coverage in children aged 3, 4, and 5 months were very high in China in 2019 (99%), pertussis protection in terms of the prevalence of children in the target vaccination population with vaccine-induced pertussis protection (83.8%) was not sufficient to establish herd immunity and block the transmission of *B. pertussis* with values of Ro ≥ 10 in the community [23]. Since pertussis is common in adolescents, a booster immunization in the age group is recommended in many industrial countries. In response to nationwide outbreaks in 2012, maternal immunization with the tetanus, diphtheria, and acellular pertussis (Tdap) vaccine was introduced in the UK and USA [24]. The use of maternal immunization clearly provides protection to infants who are too young to be vaccinated [25,26,27]. At present, several countries with a high incidence of pertussis have implemented the strategy of maternal immunizations [28]. However, so far, no booster immunizations in China have been recommended [29].

## 5. Conclusions

Our results showed that, in China, about two-thirds of adults at childbearing age and more than 80% of infants before the age of primary pertussis immunizations did not have pertussis-specific antibodies, suggesting that they were vulnerable to pertussis. To protect against pertussis in these populations, booster vaccinations in adults should be considered in China.

## Figures and Tables

**Figure 1 vaccines-10-00084-f001:**
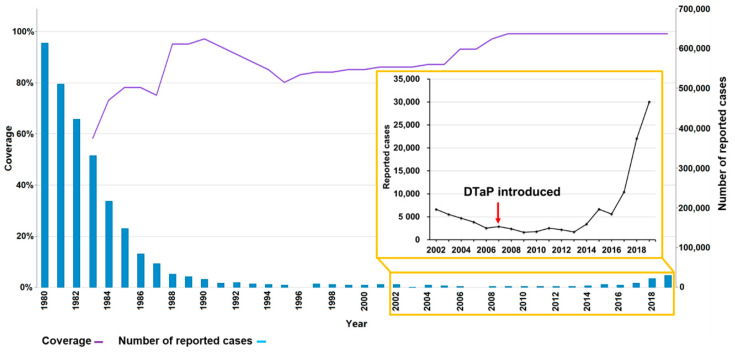
DTP3 vaccination coverage and number of reported pertussis cases in China, 1980–2019 [2,3]. DTP3: diphtheria and tetanus toxoids and pertussis (DTP)-containing vaccine, 3rd dose. This is an adaptation of an original work “Comparison of Immunization coverage for Diphtheria Tetanus Toxoid and Pertussis (DTP) vaccination coverage and Reported cases for Pertussis. Geneva: World Health Organization (WHO); 2021. Licence: CC BY-NC-SA 3.0 IGO”.

**Figure 2 vaccines-10-00084-f002:**
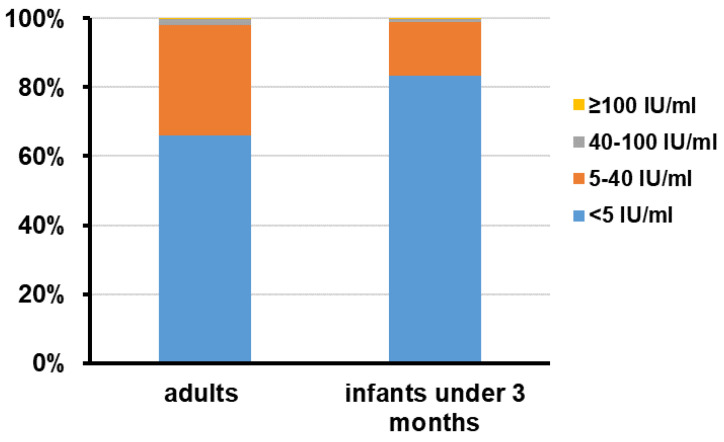
Distribution of serum PT IgG antibodies concentrations in adults at childbearing age and infants under 3 months old. The number of serum specimens with PT IgG antibodies concentrations ≥ 100 IU/mL, 40~100 IU/mL, 5~40 IU/mL, and <5 IU/mL in each group was calculated and the data are shown as percentage.

**Figure 3 vaccines-10-00084-f003:**
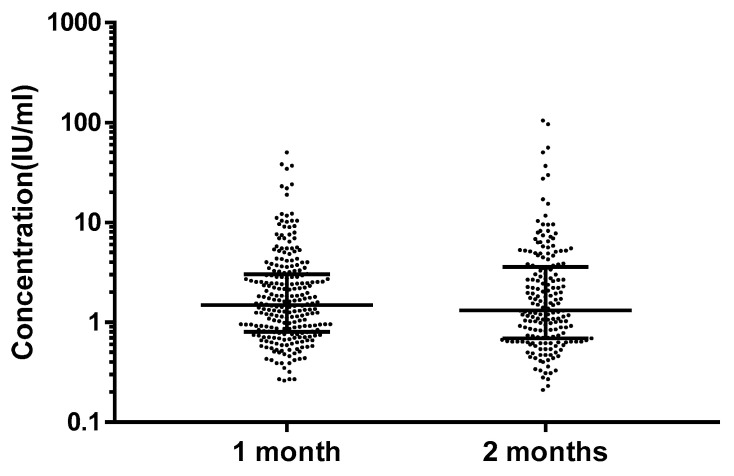
Serum PT IgG antibodies concentrations of one-month-old and two-month-old infants. The data are shown as median ± interquartile range. No statistical difference was observed between the two groups (*p* = 0.56).

**Table 1 vaccines-10-00084-t001:** Characteristics of 1175 study subjects and their serum anti-PT IgG antibodies.

Characteristics	Infants ^1^ No (%)	Adults ^2^ No (%)	*p* Value
Total	440	735	
Gender			
Male	222 (50.45)	407 (55.37)	
Female	218 (49.55)	328 (44.63)	
Seroprevalence ^3^			
Total	5 (1.14)	15 (2.04)	0.25
Male	3 (1.35)	10 (2.46)	
Female	2 (0.92)	5 (1.52)	
Proportion of subjects with non-detectable anti-PT IgG ^4^			
Total	367 (83.41)	485 (65.99)	<0.001
Male	181 (81.53)	264 (64.86)	
Female	186 (85.32)	221 (67.38)	
Anti-PT IgG concentration (IU/mL)			
Geometric mean (IU/mL)			
Total	1.72	3.81	
Male	1.78	3.84	
Female	1.65	3.78	
Median (IU/mL)			
Total	1.43	3.24	
Male	1.47	3.27	
Female	1.41	3.18	
Range (IU/mL)			
Total	0.21–104.56	0.52–186.50	
Male	0.26–104.56	0.53–186.5	
Female	0.21–50.24	0.52–81.23	

^1^ Young infants under 3 months old; ^2^ Adults at childbearing age; ^3^ Proportion of subjects with anti-PT IgG ≥ 40 IU/mL; ^4^ A cut-off < 5 IU/mL indicated that the antibodies were non-detectable. PT: pertussis toxin.

## Data Availability

Reported pertussis cases and DTP3 coverage in China during 1980–2019 were summarized and visualized according to data from the WHO Immunization data pool at https://immunizationdata.who.int/compare.html?COMPARISON=type1__WIISE/MT_AD_COV_LONG+type2__WIISE/MT_AD_INC_LONG+option1__DTP_coverage+option2__PERTUSSIS_cases&CODE=CHN&YEAR= (accessed on 13 September 2021), and the National Health Commission of the People’s Republic of China at http://www.nhc.gov.cn/jkj/s7923/202108/7337fd75c8b749309a2de28aec1a03bd.shtml (accessed on 13 September 2021). Other data presented in this study are available on request from the corresponding author. The data are not publicly available due to intellectual property considerations.

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
