# Peer review of "Seroprevalence Study of Pertussis in Adults at Childbearing Age and Young Infants Reveals the Necessity of Booster Immunizations in Adults in China"

_vaccines, 2022, doi:10.3390/vaccines10010084_

Round 1

Reviewer 1 Report

This study uses serology to measure the levels of anti-pertussis toxin IgG in sera taken from adults aged 20-39 and infants under 3 months of age in 2019 in Beijing, China. The authors found that approximately 2/3 of adults and 80% of infants had undetectable levels of anti-pertussis toxin IgG which may indicate that they are susceptible to pertussis infection. They recommend the introduction of adult booster vaccinations.

This is similar studies performed by these authors and others which have looked at the seroprevalence of anti-PT IgG in Beijing. The results found in the present study correspond to those obtained from the other studies. This study is novel in the fact that it uses samples from 2019.

Overall, the study is an interesting seroprevalence study and gives an indication of the levels of pertussis disease in 2019. It also gives an indication of waning immunity in individuals which is important for vaccination schedules. I think the manuscript should be accepted with the following modifications.

The authors mention in the Abstract and Discussion that the samples were taken in 2019 but it should also be stated in the Materials and Methods.

While overall the manuscript is well written there are very minor grammatical errors scattered about and it would benefit from being proof read.

Line 28 – Bordetella pertussis should be in italics (italics should be used when the bacterium is mentioned in other parts of the manuscript too)

Reviewer 2 Report

The paper presents the results of a seroprevalence study carried out in China in 2019 where anti-pertussis IgG antibodies were assessed in 1175 serum samples, 735 adults aged 20-39 years and 440 infants under 3 months. The study found very low mean antibody levels and undetectable antibody levels in 66% in adults and 83.4% in infants. The conclusion of the study is that booster vaccinations in adults should be considered.

I suggest the following points to increase the quality of the paper:

  1. The Abstract must present the information on the mean or median antibody levels, and seroprevalence of antibody levels ≥ 40 IU/ml.
  2. Lines 32-35. This paragraph is confusing. It must be reviewed and rewritten.
  3. I suggest to include a comment about pertussis immunity in children aged less than one year (target vaccination population) in China in 2019, which can be found in the paper recently published in Vaccines:

Vaccination Coverage for Routine Vaccines and Herd Immunity Levels against Measles and Pertussis in the World in 2019. Vaccines. 2021; 9(3):256. https://doi.org/10.3390/vaccines9030256

The study found (Supplementary file) that percentages of DTaP vaccination coverage in children aged 3, 4 and 5 months were very high in China in 2019 (99%), but pertussis protection in terms of prevalence of children in the target vaccination population with vaccine-induced pertussis protection (83.8%) was not sufficient to establish herd immunity and block transmission of Bordetella pertussis with values of Ro ≥ 10 in the community.

  1. Lines 64-69. Indicate the year of the study. Why the informed consent was not applicable?
  2. Subheading 2.1. What information and how was collected from participants? Information on pertussis vaccination was obtained from adults? If pertussis vaccination was introduced in 1960s, many adults should have received pertussis vaccines. 
  3. Line 70. The title for 2.2 subheading is the same as for 2.1. 
  4. Lines 73-74. Include references for interpreting IgG levels ≥ 40 IU/ml as past infection and ≥ 100 IU/ml as recent infection.
  5. Lines 79-81. Indicate what was assessed, statistical tests used and p values.
  6. Results, lines 83-86. Indicate the participation rate.
  7. Table 1. Indicate No (%) below the words Infants and Adults.
  8. Table 1. Indicate what does it mean seroprevalence and undetective prevalence using notes.

Reviewer 3 Report

Congratulations, an interesting study on the seroprevalence of antibodies against bortedella pertussis in two specific populations. The study shows how it is common to find a large part of the population that does not have defenses against this bacterium and that, therefore, will benefit from an extra booster of the vaccine.

As a question to the authors, I would like to ask if a DTaP vaccine booster is administered during pregnancy to try to ensure that newborns have good levels of antibodies until their vaccination is due.

Thank you for your comments

Regards

Author Response

Please see the attchment.

Round 2

Reviewer 2 Report

The revised version of the paper is correct.